# Virological Correlates of IgM–IgG Patterns of Response to SARS-CoV-2 Infection According to Targeted Antigens

**DOI:** 10.3390/v13050874

**Published:** 2021-05-10

**Authors:** Pablo Barreiro, Francisco Javier Candel, Juan Carlos Sanz, Jesús San Román, María del Mar Carretero, Marta Pérez-Abeledo, Belén Ramos, José Manuel Viñuela-Prieto, Jesús Canora, Francisco Javier Martínez-Peromingo, Antonio Zapatero

**Affiliations:** 1Public Health Regional Laboratory, Hospital Isabel Zendal, Av. Manuel Fraga Iribarne, 2, 28055 Madrid, Spain; franciscojavier.candel@salud.madrid.org (F.J.C.); juan.sanz@salud.madrid.org (J.C.S.); jesus.sanroman@urjc.es (J.S.R.); maria.carretero@salud.madrid.org (M.d.M.C.); mpabeledo@salud.madrid.org (M.P.-A.); belen.ramos@salud.madrid.org (B.R.); jovinuel@ucm.es (J.M.V.-P.); 2Council of Public Health, Community of Madrid, Calle O’Donnell, 50, 28009 Madrid, Spain; jesus.canora@salud.madrid.org (J.C.); fmperomingo@salud.madrid.org (F.J.M.-P.); antonio.zapatero@salud.madrid.org (A.Z.)

**Keywords:** SARS-CoV-2, COVID-19, serology, IgM, spike, nucleocapsid

## Abstract

The virological meaning of the different patterns of serology in COVID-19 has been little examined in clinical settings. Asymptomatic subjects with IgM-spike (S) and IgG-nucleocapsid (N) determinations by chemiluminescence were studied for SARS-CoV-2 shedding in respiratory secretions by transcription-mediated amplification (TMA). In subjects showing IgM-S positive and IgG-N negative, IgG-S was determined by lateral flow assay. A total of 712 individuals were tested: 30.0% presented IgM-S(+)/IgG-N(−), 25.8% had IgM-S(+)/IgG-N(+) and 44.2% had IgM-S(−)/IgG-N(+); the proportion with TMA(+) were comparable in these three groups: 12.1, 8.7 and 10.5%, respectively. In individuals with IgM-S(+)/IgG-N(−), IgG-S(+) was detected in 66.5%. The frequency of IgM-S(+)/IgG-S(−) in the total population was 10.0%, of whom 24.1% had TMA(+); the chances for TMA(+) in subjects with an IgM-S(+) alone pattern were 2.4%. Targeting of the same SARS-CoV-2 antigen seems to be better for the characterization of IgM/IgG patterns of response. IgM-S(+) alone reactivity is rare, and a small proportion is associated with viral shedding.

## 1. Introduction

Direct diagnosis of severe acute respiratory syndrome coronavirus 2 (SARS-CoV-2) is today based on antigen and molecular tests [1]. Serology assays provide additional information, mostly of epidemiological interest as to determine the prevalence of infection in special groups [2] or monitor the evolution of the pandemic in the general population [3]. Although humoral immunity has little implication during the initial course of primary infection, the presence of neutralizing antibodies after coronavirus disease (COVID-19) is related with protection from future reinfections [4,5,6,7,8]. Serological studies may help with indicating or monitoring the efficacy of a SARS-CoV-2 vaccine [9].

Classically, the IgM and IgG responses have been used to determine the different phases of viral infections, namely acute, convalescent or chronic. Guidelines indicate that subjects with IgM reactivity but with negative IgG should be tested to discard active SARS-CoV-2 infection [10,11], although IgM and IgG arouse mostly at the same time after first infection [12,13]. Additionally, the SARS-CoV-2 nucleocapsid (N) and spike (S) proteins have different kinetics, as anti-N antibodies decline earlier than anti-S [14,15,16,17]. Little is known about the correlation between the kinetics of IgM and IgG and SARS-CoV-2 shedding in respiratory secretions. Herein we have evaluated the virological correlates, as the presence of SARS-CoV-2 RNA in nasal and pharyngeal swabs, in patients with the three possible IgM and IgG patterns.

## 2. Methodology

### 2.1. Participants

The seroSOS study is a SARS-CoV-2 serological point survey of most sociosanitary centers in the Community of Madrid. All asymptomatic residents and workers in these centers were offered to participate with a serology in blood and a nasopharyngeal swab as a molecular test for SARS-CoV-2. We selected for this study all consecutive participants for whom IgM plus IgG serology was available.

### 2.2. Microbiological Assays

The Abbott^®^ SARS-CoV-2 assay, that detects anti-S IgM (IgM-S) and anti-N IgG (IgG-N) by chemiluminescent microparticle immunoassay (CMIA) with peak sensitivities of 96 and 99.6%, respectively [18], was used in all subjects. In individuals showing IgM-S-positive and IgG-N-negative reactivity, a second serology test that detected both IgM-S and IgG-S antibodies with a positive percent agreement of 85–88% (lateral flow immunochromatographic assay (LFIA) by Autobio^®^ Diagnostics Co., Zhengzhou, China [19]) was used.

Detection of SARS-CoV-2 RNA from nasopharyngeal swabs was undertaken by transcription-mediated amplification (TMA, Panther^®^, Hologic-Grifols^®^) [20].

### 2.3. Statistical Analysis

Clinical information was recorded in electronic databases (Forms^®^, Microsoft^®^). Comparison of proportions was done by Chi-2, with Yates’ correction if needed, with Epidat^®^ software (version 4.2, July 2016). The Regional Ethics and Investigation Committee of the Community of Madrid approved the study.

## 3. Results

Serum samples from 712 asymptomatic individuals were available for the determination of IgM-S and IgG-N responses. A total of 449 subjects were identified as residents (68.9% women, mean age 81.2 ± 14.8 years old) and 263 subjects as workers (84.9% women, mean age 41.5 ± 12.0 years-old) in nursing homes. History of past COVID-19 was registered in medical records in 156 (35.3%) residents and 68 (26.3%) workers (*p* = 0.007).

Positive IgM-S was found in 439 (61.7%) and positive IgG-N in 498 (69.9%) individuals. Patterns of antibody response were as follows: IgM-S positive plus IgG-N negative in 214 (30.0%) subjects, IgM-S positive plus IgG-N positive in 184 (25.8%) subjects and IgM-S negative plus IgG-N positive in 314 (44.1%) (Table 1).

Nasopharyngeal swabs, all sampled at the time of blood test, were available for 688 individuals, 407 residents (90.6%) and 241 workers (91.6%). Positive TMA was detected in 72 (10.5%) individuals, similarly distributed among 56 (12.5%) residents and 24 (9.1%) workers. Molecular tests were available in 149 subjects showing an IgM-S positive alone pattern, in 184 subjects with both positive IgM-S and IgG-N, and in 314 subjects with an IgG-N positive alone pattern. No significant differences in terms of proportion of individuals with positive TMA were found among those three serological patterns, 12.1% for IgM-S positive alone, 8.7% for IgM-S plus IgG-N positive and 10.5% for IgG-N positive alone (Figure 1). History of COVID-19 was confirmed in 52.4% of subjects with IgM-S positive alone, in 46.1% of subjects with positive IgM-S and IgG-N and in 49.5% of subjects with IgG-N positive alone (*p* = NS).

A second antibody test by LFIA to detect IgM-S plus IgG-S was performed in serum samples from 206 patients with IgM-S positive alone by CMIA. In 91.3% cases the second IgM-S result was negative (hypothetically false negative), while in 66.5% cases the second IgG-S result was positive (hypothetically true positive). The chances for detection of positive TMA were 24.1% among IgM positive plus IgG negative for samples tested against S antigen, while this figure was 12.1% if IgM against S but IgG against N were used (*p* = 0.03) (Figure 1).

## 4. Discussion

Although serology tests for viral infections mostly inform of past or chronic infection, detection of IgM reactivity with negative IgG sets an indication to discard active infection [10,11]. Our analyses have shown that the chances of detecting SARS-CoV-2 shedding are low (12%) in individuals with an IgM positive alone response.

Lateral flow rapid test (Autobio^®^) failed to detect IgM-S in more than 90% of subjects that were IgM-S positive alone by CMIA. Previous studies have shown that, even in patients with history of positive PCR for SARS-CoV-2, colloidal-gold immune assays have much lower accuracy for IgM-S (sensitivity of 47%) [21] when compared with IgM-S as detected by CMIA (sensitivity of 86%) [22]. The IgM-S positive plus IgG-N negative pattern was observed in nearly one-third of subjects with SARS-CoV-2 seropositivity. However, when IgG was targeted against S, up to two-thirds resulted IgG-S positive, rendering a true IgM-S positive alone seroresponse as low as 10%. Among subjects with exposure to SARS-CoV-2, the probability of detecting a positive nasopharyngeal swab based on IgM-S positive IgG-S negative was 2.4%. Given that IgM is not detected substantially earlier than IgG, the significance of this positivity of molecular tests may be put under question [12,23,24].

Differences in the dynamics of S versus N antibodies may in part account for the discrepancies observed in our study with respect to IgG responses. A rapid decline in IgG-N titers has been shown previously, while levels of IgG-S remain detectable for longer [25]; the mean half-life of IgG-N is 53 days, while for IgG-S it reaches 81 days, according to a recent study [26]. Other factors such as age, ethnicity or the severity of infection may affect the waning of SARS-CoV-2 antibodies [27,28,29]. The use of SARS-CoV-2 IgG for seroepidemiological studies recommends targeting S rather than N antigens to ensure detection of memory responses for longer periods of time [27,30,31,32].

We feel that the failure of the N-based IgG test to detect memory response is supported by the fact that viral shedding was twice as frequent in IgM-S positive plus IgG-S negative (24.1%) than in IgM-S positive plus IgG-N negative samples (12.1%). A recent study showed that S-based serology was able to detect IgG activity in up to 69% of subjects with negative IgG against N antigen [33]. If IgM and IgG are both directed against the S, a significant number of false IgG-N negative results is avoided (as they result IgG-S positive) and fewer IgM-S positive alone (with IgG-S negative) results are detected; therefore, samples that were truly in the initial phases of the infection. Differences in the methods and accuracy of techniques used to detect IgG-N (CMIA) and IgG-S (LFIA) are acknowledged as a limitation of the study.

Targeting S antigens better reflects neutralizing immune activity and may better indicate immune protection than N antigens [34,35]. Additionally, S-based seroresponse is needed to monitor vaccination efficacy [36]. The ability to quantify IgG-S levels, as second generation high-throughput serological assays display, may add information about the intensity of immune response after natural infection or vaccination [37]. Still, N antigens are better suited to discriminate immunity from infection rather than from vaccination.

In conclusion, our analyses support the utility of targeting S-protein for the monitoring of antibody response against SARS-CoV-2. The use of IgM is of little interest given that both IgM and IgG have a very similar kinetics. Targeting N-protein is needed to establish that immunity was acquired after natural infection rather than just by vaccination.

## Figures and Tables

**Figure 1 viruses-13-00874-f001:**
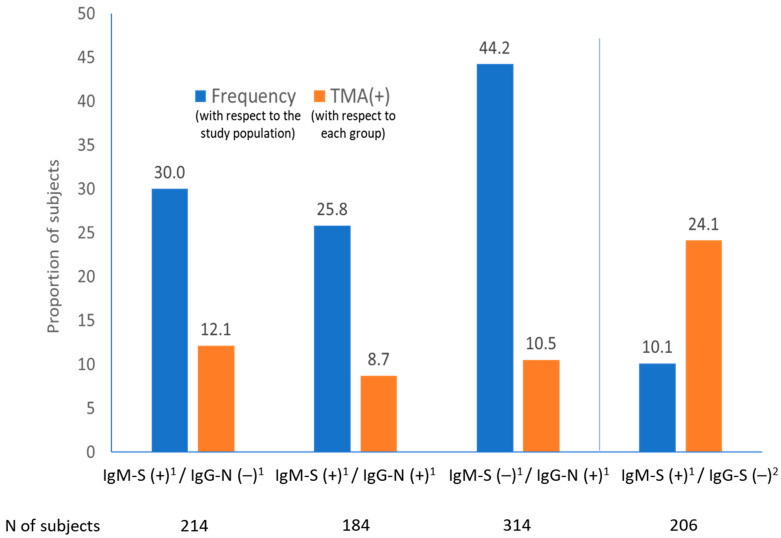
Proportion of subjects with SARS-CoV-2 shedding (TMA+) according to type of serological response and targeted antigens. S, spike protein; N, nucleocapsid protein. (1) IgM-S and IgG-N were done by chemiluminescence immunoassay method in the Abbott^®^ Architect^®^ platform. (2) IgG-S tests were done by rapid lateral flow immunochromatography by Autobio^®^. Transcription-mediated amplification (TMA) was done by Panther^®^, Hologic-Grifols^®^. N, nucleocapsid; S, spike.

**Table 1 viruses-13-00874-t001:** Distribution of type of antibody reaction in venous blood samples by CMIA (Architect^®^, Abbott^®^) from subjects with exposure to SARS-CoV-2.

	All (%)	Residents (%)	Workers (%)	*p* Value
No.	712	449 (63.0)	263 (37.0)	
IgM-S positive	439 (61.7)	229 (51.0)	164 (62.4)	0.004
IgG-N positive	498 (69.9)	338 (75.3)	167 (63.5)	<0.001
Patterns:				
IgM-S positive/IgG-N negative	214 (30.0)	108 (24.0)	95 (36.1)	0.001
IgM-S positive/IgG-N positive	184 (25.8)	120 (26.8)	69 (26.2)	NS
IgM-S negative/IgG-N positive	314 (44.2)	221 (49.2)	99 (37.7)	0.002

## Data Availability

The data presented in this study are available on request from the corresponding author. The data are not publicly available due to being still under analysis.

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
