# Peer review of "Virological Correlates of IgM–IgG Patterns of Response to SARS-CoV-2 Infection According to Targeted Antigens"

_viruses, 2021, doi:10.3390/v13050874_

Round 1

Reviewer 1 Report

This study is for the serologic pattern of 712 subjects asymptomatic residents of workers (12 subjects are unknown?) in Madrid. Your manuscript is hard to understand because the number of results seems to be not concordant. There are other results that make readers confusing. For example, secondary antibody test was performed 206 samples which firstly showed showing the IgM-S (+) / IgG-N (-), but you fused graph in the Fig 1.

Author Response

Dear reviewer, thanks for your comments. Yes, there were 12 subjects with no information about their being worker or resident. We have clarified their condition and correct non-matching figures. The figure has also been clarified as requested. Many thanks

Reviewer 2 Report

This is an interesting article regarding Virological Correlates of IgM-IgG Patterns of Response to 2 SARS-CoV-2 Infection.

The manuscript is interesting and well developed. 

I have some minor comments:

  • Authors should add the clinical implications of their article
  • The references should be improved: I suggest COVIDSurg Collaborative. Delaying surgery for patients with a previous SARS-CoV-2 infection. Br J Surg. 2020 Nov;107(12):e601-e602. doi: 10.1002/bjs.12050; Bellato V, Konishi T, Pellino G, An Y, Piciocchi A, Sensi B, Siragusa L, Khanna K, Pirozzi BM, Franceschilli M, Campanelli M, Efetov S, Sica GS; S-COVID Collaborative Group. Impact of asymptomatic COVID-19 patients in global surgical practice during the COVID-19 pandemic. Br J Surg. 2020 Sep;107(10):e364-e365. doi: 10.1002/bjs.11800; Lima DS, Ribeiro MAF Jr, Gallo G, Di Saverio S. Role of chest CT in patients with acute abdomen during the COVID-19 era. Br J Surg. 2020 Jun;107(7):e196. doi: 10.1002/bjs.11664. Epub 2020 May 9
  • The limitations of the study should be added
  • English Editing

Author Response

Dear reviewer, clinical implications, limitations of the study were added and references have been improved. Many thanks,

Reviewer 3 Report

The authors evaluated the virological correlates, as the presence of SARS-CoV-2 RNA in nasal and pharyngeal swabs, in patients with the three possible IgM and IgG patterns that indicate COVID-19. The data support the utility of targeting S-protein for the monitoring of antibody response against SARS-CoV-2. IgM's use has probably little interest given that both IgM and IgG have very similar kinetics, making the IgM positive alone pattern challenging to detect and with little correlate with SARS-CoV-2 shedding. It is a fascinating study for COVID-19 related study, and I also have some minor suggestions for the authors.

1, Line 99, 100, 104: ”vs” should be corrected as ”vs. ”, the author needs to use a consistent format for the whole manuscript. I suggest that they can take a look at the recent publication in “Viruses” (Figure 2)  https://www.mdpi.com/1999-4915/13/4/600/htm

2, Line 169: ……was observed in nearly” one third” of…….., ” one third” should be corrected as” one-third”

Author Response

Dear reviewer, format of te text has been improved as suggested. Many thanks

Round 2

Reviewer 1 Report

I have checked your totally revised manuscript. The authors has insisted the viral shedding was twice as frequent in IgM-S positive alone. It is about the detection of acute SARS-CoV-2 infection. But the conclusion told about the utility of targeting S-protein for monitoring of antibody response against SARS-CoV-2. Also your results was derived from the insensitive assay (LFIA) false negatives. There are many possibilities for this result including the performance of assays. You have to describe for that limitations (e.g. sensitivity and specificity of serological assays used). Some minor points are below.

line 111 The corrected number of 30.1% have to be add into table and figure.

line 115 Table 2 -> Table. There are loss of title only in sociosanitary centers.

line 122 Genetic -> Molecular

line 139 (Figure 2 -> Figure

first number in the Figure 30.0 -> 30.1

line 140 Correction of the legend of figure is needed.

lline 142 Is this legend or manuscript? (same with method)

Author Response

                                                                                              Madrid, May 4th 2021

Dear Editor and Reviewers,

Please find here the response to the queries risen by the referees. We wish to acknowledge their contribution to improve the quality of the paper.

Virological Correlates of IgM-IgG Patterns of Response to SARS-CoV-2 Infection According to Targeted Antigens

Pablo Barreiro1,2 *, Francisco Javier Candel1,2, Juan Carlos Sanz1, Jesús San Román1,2, María del Mar Carretero1,2, Marta Pérez-Abeledo1, Belén Ramos1, José Manuel Viñuela1, Jesús Canora2, Francisco Javier Martínez-Peromingo2 and Antonio Zapatero2

1Public Health Regional Laboratory and 2Council of Public Health, Community of Madrid.

Reviewer 1:

The authors have insisted the viral shedding was twice as frequent in IgM-S positive alone. It is about the detection of acute SARS-CoV-2 infection. But the conclusion told about the utility of targeting S-protein for monitoring of antibody response against SARS-CoV-2. Also your results was derived from the insensitive assay (LFIA) false negatives. There are many possibilities for this result including the performance of assays. You have to describe for these limitations (e.g. sensitivity and specificity of serological assays used):

The probability of having positive TMA was twice greater for IgM-S pos / IgG-S neg than for IgM-S pos and IgG-N neg. In subjects with IgM-S pos and IgG-N neg by CMIA, two-thirds are IgM-S pos and IgG-S pos by LFIA (despite lower sensitivity of this technique). To us this means that if you test IgM and IgG against the spike, you avoid a large number of false IgG-N negatives, and you truly detect IgM positives alone, therefore in the initial phases of the infection. This could be the reason why targeting the spike is more sensitive for detecting early viral shedding. The issue of lower sensitivity of LFIA versus CMIA favors our results, as we where able to detect IgG-S positives by LFIA (less sensitive) when they were IgG-N negatives by CMIA (more sensitive). We have better explained this hypothesis in the Discussion and included sensitivity and specificity of both techniques in Methodology. We have included disparities between CMIA and LFIA as a limitation of the study.

Line 111 The corrected number of 30.1% have to be add into table and figure:

Correct number is 30.0% (all three proportions need to add to 100%) and has been included in text, table and figure.

Line 115 Table 2 -> Table. There are loss of title only in sociosanitary centers.

This mistake has been corrected.

Line 122 Genetic -> Molecular

This mistake has been corrected.

Line 139 (Figure 2 -> Figure

This mistake has been corrected.

First number in the Figure 30.0 -> 30.1

This mistake has been corrected. Correct number is 30.0%, decimals are rounded so that all three proportions (30.0%, 25.8% and 44.2%) add to 100% as needed.

Line 140 Correction of the legend of figure is needed.

Legend has been corrected

Line 142 Is this legend or manuscript? (same with method)

It is part of the manuscript